# Experimental Investigation of Milling Performance of Silicon Nitride Ceramic Subject to Different Assisted Systems

**DOI:** 10.3390/ma16010137

**Published:** 2022-12-23

**Authors:** Muhammad Naveed Raza, Shen-Yung Lin

**Affiliations:** Department of Mechanical and Computer-Aided Engineering, National Formosa University, 64, Wunhua Rd., Huwei Township, Yunlin County 632, Taiwan

**Keywords:** silicon nitride, laser-assisted, hybrid-assisted, Taguchi method, milling

## Abstract

In this study, silicon nitride milling experiments are carried out using Polycrystalline Diamond (PCD) end mill rods under unassisted, hybrid-assisted (combination of laser assisted and three axis ultrasound), and laser-assisted systems to examine the cutting performance and machined surface quality of different cutting tools. The best combination of process parameters for silicon nitride composites milling are obtained using the Taguchi method. The effects of spindle speed, radial depth of cut, and feed rate on surface roughness, cutting force, edge topography, and tool wear of silicon nitride surfaces are investigated. The results reveal that hybrid-assisted produces superior surface roughness, longer tool life, fewer machining defects, and lower cutting force than unassisted. Best results of triaxial ultrasonic-assisted combined with laser on cutting performance are achieved as the ultrasonic waves help to vibrate the cutting tool and workpiece simultaneously, which helps to effectively remove chips and lowers the cutting force. When compared to unassisted milling, laser-assisted and hybrid-assisted milling improve total average surface roughness by 42% and 66%, and total cutting forces by 26% and 46%, respectively. The best processing parameters obtained in this study are high spindle speed (12,000 rpm), low feed rate (500 mm/min), and low cutting depth (0.02 mm).

## 1. Introduction

The choice of a material (such as glass, ceramic, metal, alloy, etc.) for a potential application is made based on the different properties of that material [1,2,3]. Lot of research has been conducted to evaluate the mechanical, structural, and other properties of various materials under certain testing conditions and environments. In the similar way, silicon nitride has been widely tested as it is a promising advanced engineering ceramic due to its superior properties such as high strength at elevated temperatures, low thermal conductivity, corrosion resistance, excellent wear resistance, high fracture toughness, high strength to weight ratio, good flexural strength, and low-density [4,5]. Silicon nitride has been widely used in aerospace, national defense, biomedical, and other critical industrial domains due to these qualities [6,7,8,9]. Some other potential applications and detailed properties and fabrication methods of silicon nitride can be seen in literature [7,10,11]. The material may be overly hard and brittle during the cutting process, causing the silicon nitride edge to be readily broken [4,5]. In terms of literature review, Guerrini et al. [12] investigated the impact of a diode laser with a wavelength of 980 nm on the single pass grinding performance of silicon nitride with an ASD150-R75B99-1/4 grinding wheel. The results demonstrated the procedure’ efficacy in terms of minimal thermal cracking, which resulted in consistent and precise cracks. These consistent and precise cracks decreased the grinding pressures and the extended the tool life.

A group of authors designed laser-assisted milling (LAM) to explore the cutting process of sintered silicon nitride [13]. Cutting forces, chip morphology, surface roughness, and subsurface damage were studied as a function of operation parameters. The findings demonstrated the viability and benefits of LAM in ceramics, including a smooth surface, reduced cutting force and tool wear, and no microcrack zones [13]. Melkote [14] created a unique laser-assisted micro-milling machine that allows freeform three-dimensional micro-scale shapes to be created in hard materials. The hardened A2 tool steel workpiece was used in experiments. When compared to traditional planning, Chang Chih-Wei and Chun-Pao Kuo [15] claimed that LAM significantly reduced the cutting force by 22% (feed force) and 20% (push force). The surface integrity of the workpiece was better in the LAM process than in traditional planning. Budong Yang and Shuting Lei [16] investigated the machinability of Si_3_N_4_ under laser-assisted milling by evaluating tool wear and workpiece edge chipping. LAM was found to considerably increase the machinability of Si_3_N_4_ by lowering cutting pressures and extending tool life. Patten et al. [17] used a single-point cutting tool to approximate orthogonal silicon nitride cutting. Theoretically, ductile machining of silicon nitride was conceivable with a short tip radius, fast speeds, and small feeds, favored by positive or zero-degree rake angles. To predict cutting force and temperature effects, Shen et al. [18] developed a finite element model for laser-assisted milling of silicon nitride. A parametric research also considers other variables such as rake angle, depth of cut, local damping coefficient, and cluster size. It demonstrates that in laser-assisted machining, all of these variables affect surface/subsurface cracks and chip development in silicon nitride ceramics [18,19]. Tian et al. [20] studied the laser-assisted milling of silicon nitride. They demonstrated that laser-assisted milling reduces tool wear, improves surface smoothness, and improves repeatability. They demonstrated experimentally that as temperature rises, surface roughness and tool wear decrease. Kim and Lee [21] used a newly designed back-and-forth preheating strategy to investigate laser-assisted ball-end milling of difficult-to-cut materials. The proposed back-and-forth preheating method in laser-assisted milling on Inconel 718, zirconia, and silicon nitride resulted in decreased cutting force and improved surface quality. Kang and Lee [22] investigated laser-assisted milling for turning operations on silicon nitride and employed a back-and-forth preheating method to achieve the requisite temperature. The proposed technique and constitutive equation can help in laser-assisted milling of ceramics.

The pertinent literature review suggests that the majority of cutting processing and numerical simulation research on hard and brittle composite materials such as silicon nitride ceramic has been conducted in recent years using traditional milling as well as ultrasonic and laser-assisted milling. Hard and brittle materials subjected to laser-assisted systems have been studied extensively during the past few years [23,24]. For example, a comprehensive review related to the pros and cons of laser-assisted machining of titanium alloys can be found in the literature [24]. In these review works, machining qualities and optimization of different kinds of laser-assisted systems subjected to various hard and brittle materials are summarized. In another study [25], the influence of cutting zone temperature and vibration on tool wear and surface roughness has been discussed. In order to provide a general perception of tool/surface finish interaction, a study reports the residual stresses and surface finish resulted by dry machining (such as milling, drilling, and turning) of AA7075-T651 [26]. The effects of cutting tool, cutting speed, and feed rate on the surface finish and flank wear during turning of AISI 304 steel was the topic of investigation in another study [27]. The authors have reported that feed rate was the most influential factor for both flank wear and surface roughness. Similar to laser-assisted, ultrasonic-assisted systems have been investigated by some researchers [28,29,30]. However, to the authors’ best knowledge, no study has been found in the literature that reports the combined effect of a laser and ultrasonic-assisted system on the milling performance of silicon nitride ceramic. In fact, this technology is still in its infancy.

Keeping in view the highlighted research gap, the prime novelty of this study is to use a hybrid-assisted (combination of laser-assisted and three axis ultrasound) system with the existing cutting tools for the milling experiments in order to improve tool life and surface roughness while significantly reducing the cutting force. According to the literature review, milling is still a viable processing method for this composite material because they it remove excess material from the mold near the net shape product. This study uses PCD end mill rods to execute silicon nitride milling trials under unaided, hybrid-assisted (laser-assisted and three axis ultrasound), and laser-assisted systems to evaluate the cutting performance and machined surface quality of various cutting-tools. For milling silicon nitride composites, the Taguchi method’s smaller-is-better characteristics are used to determine the ideal set of process parameters. The objective function, subjecting constraint, and less significant cutting tool wear are, respectively, machined surface roughness and machined defects close to the material’s edge. On the surface roughness, cutting force, edge topography, and tool wear of silicon nitride surfaces, the impacts of spindle speed, radial depth of cut, and feed rate are examined.

## 2. Materials and Methods

### 2.1. Materials

The material utilized in the experiment is silicon nitride with dimensions of 70 mm × 70 mm × 2 mm; its appearance is shown in Figure 1a. The corresponding mechanical and physical properties are shown in Table 1. The experiment uses a polycrystalline diamond (PCD) milling cutter from the Taiwan Diamond Company (Taoyuan City, Taiwan). This can be seen in Figure 1b. PCD milling cutter’s effective blade length and shank and outer diameters are all 6 mm. The back-and-forth method is used for preheating to achieve required temperature for machining of hard and brittle materials [21]. In this method, the tool is stopped during preheating, and then the laser spot and tool are moved together after preheating. Preheating is done without a tool four times and with a tool the fifth time in this experiment. The back-and-forth preheating method has the advantage of increasing the laser and tool spot temperatures. Furthermore, when compared to preheating, the oxidation zone can be lowered by simply increasing the laser power, as can be seen in Figure 1c.

### 2.2. Experimental Setup

A TC-500 processing machine is used for the experiment, as shown in Figure 2. A dynamometer (Kistler type 9257B power meter, Winterthur, Switzerland) is used to measure the cutting force. The ultrasonic spindle module provided by Spintech Company (Taichung, Taiwan) in the experiment includes an ultrasonic driver, an ultrasonic tool holder, and a laser displacement meter. The tool amplitude is measured by laser displacement measurement, and the voltage is adjusted through the ultrasonic driver to control the tool amplitude value. Under hybrid-assisted cutting (three-axis ultrasound plus laser-assisted), the biaxial ultrasonic oscillation platform oscillator produces 25.7 kHz oscillation frequency with an amplitude of 2~2.5 μm; the ultrasonic tool holder produces 25.3 kHz oscillation frequency with an amplitude of 2.5 μm, since the biaxial ultrasonic oscillation platform is the largest, but it will be reduced when the silicon nitride is held through a jig. The actual amplitude of the workpiece measured by laser displacement measurement is 0.7~0.9 μm. During the experiment, a 220-watt high-power carbon dioxide laser was used to preheat for laser-assisted and hybrid-assisted silicon nitride grinding and milling. The back-and-forth approach is used for pre-heating and softening of silicon nitride. The workpiece is moved without the use of a cutting tool in this manner. To achieve this high temperature, the laser was utilized twice using a back-and-forth method and once with a cutting tool instrument. After many tests, the dual-axis laser power is set at 80%, which is about 176 watts. The average temperature is about 890 °C to 900 °C.

### 2.3. Experimental Procedure

In this study, experiments were carried out for end milling of silicon nitride with unassisted, laser-assisted, and hybrid-assisted systems. Hybrid-assisted is combination of laser-assisted and three axis ultrasound. Cutting performance of hybrid-assisted, laser-assisted, and unassisted cutting was investigated. During the experiment, the surface roughness, cutting force, edge morphology, and PCD end milling cutter were recoded and observed, and a series of results were analyzed and discussed based on experimental data. This experiment plan is to discuss three kinds of assistance, namely unassisted, laser-assisted, and hybrid-assisted (laser-assisted plus three axis ultrasound). The parameters used in this experiment are shown in Table 2. Although the diameters of the two grinding rods differ, they share the same cutting speed, radial depth of cut, and feed rate. The L9 orthogonal table experiments were carried out with parameters in Table 2 plus three kinds of auxiliary and one kind of cutter, as shown in Table 3. The spindle speeds in this experiment are 6000 rpm, 9000 rpm, and 12,000 rpm, respectively; the radial depth of cut was 0.02 mm, 0.06 mm, and 0.10 mm; and the feed rate was 500 mm/min, 1000 mm/min, and 1500 mm/min. Every experiment is repeated three times. The average cutting force and surface roughness are calculated after the three experiments. The Taguchi method’s smaller-is-better characteristics are used to find the best combination of process parameters for silicon nitride composites milling, with the objective function and subjective constraints being machined surface roughness and machined defects near the material’s edge, and the less significant cutting tool wear.

## 3. Results and Discussion

This experimental strategy is based on Taguchi’s L9 orthogonal table approach. The experiments are focused on silicon nitride end milling. This study looks at three types of assistance: unassisted, laser-assisted, and hybrid-assisted (laser-assisted plus three axis ultrasound) systems. Each experiment is repeated three times, resulting in a total of 81 milling tests using three distinct types of aid. The effects of spindle speed, radial depth of cut, and feed rate on surface roughness, cutting force, edge topography of silicon nitride surfaces, and tool wear are discussed in this section. Simultaneously, the overall performance of various assistants in silicon nitride milling are compared.

### 3.1. Surface Roughness Analysis

Due to the brittleness, high hardness, and low thermal expansion coefficient of silicon nitride composites, a variety of issues might arise during the process, which is a significant factor affecting processing performance. The surface quality of machining is also affected by the three separate cutting processes. The L9 orthogonal table is utilized in this experiment to perform different assisted method on silicon nitride milling and grinding operations. The impact of various machining parameters, tools, and auxiliaries on silicon nitride surface roughness was discussed. Each data set was subjected to a variation analysis (ANOVA) to determine the contribution rate of each element and its impact on surface roughness.

Once the silicon nitride milling experiments are completed, the experimental rods are properly cleaned and subject to the surface roughness meter to discuss the influence of different processing parameters and auxiliary on the silicon nitride surface roughness. Then, analysis was performed on the variance of each data by ANOVA to check the influence and contribution of each factor to the surface roughness of the silicon nitride. The corresponding contribution rate and the factor response table are shown in Table 4, Table 5 and Table 6, and Figure 3. The contribution rate of each factor of milling under the three kinds of assistance is different. It is evident that feed rate has the highest contribution amongst the three parameters (such as spindle speed, feed rate, and depth of cut) for all three (unassisted, laser-assisted, and hybrid-assisted systems) systems. Apparently, for less feed rate, we observe less wear on the materials’ surface. It can be seen that with these three kinds of assistance, the most influential factor affecting the surface roughness is the feed rate, followed by the radial depth of cut, and finally the rotational speed.

Compared with the three aids in Figure 4, it can be found that the surface roughness values of milling of the hybrid-assisted are lower and the surface roughness of the unassisted surface is obviously higher, while surface roughness of laser-assisted is lower than unassisted but higher than hybrid-assisted. It can be inferred that under dual-axis laser-assisted irradiation and three axis ultrasounds, the surface roughness of silicon nitride is improved. Comparing nine groups of the Taguchi method, as shown in Figure 4, it can be found that surface roughness is significantly improved when feed rate and the radial depth of cutting is lowest. Figure 5 reveals that the surface increases for milling as the radial depth of cutting increases when the feed rate is fixed for all assisted cutting methods. Similarly, surface roughness increases for milling as the feed rate increases when the radial depth of cutting is fixed for all assisted cutting methods, as shown in Figure 6. Therefore, this analysis shows that the surface roughness is proportional to the feed rate and the radial depth of cut. As we see in the analysis of variance (ANOVA), the feed rate is most important factor in surface roughness followed by depth of cutting and finally the spindle speed. Under the complete Taguchi experiment, when compared to unassisted milling, laser-assisted and hybrid-assisted milling improved total average surface roughness by 42% and 66%, respectively. The optimized parameters obtained by the Taguchi method in this experiment plan are high spindle speed, low feed rate, and low radial depth of cutting.

### 3.2. Cutting Force Comparison

It can be seen from Figure 7 that the milling cutting force of the hybrid-assisted is significantly less than that of the other aids, and the unassisted is higher than that of the other aids, while the cutting forces for laser-assisted are less than the unassisted cutting forces. Therefore, the hybrid-assisted and the laser-assisted systems can reduce the cutting force as compared to unassisted. When laser power is employed, cutting forces are reduced in the laser-assisted cutting method because the hardness, shear strength, and thermo-mechanical force on the cutting tools are reduced. In the hybrid-assisted method, both laser and tri-axial ultrasonic waves are used. The ultrasonic waves help to vibrate the cutting tool and workpiece simultaneously, which helps to effectively remove chips and reduce the cutting force. Because of laser and ultrasonic waves, the cutting forces are lowest in the hybrid-assisted method compared to other assisted methods. It can be found from Figure 8 that the cutting force increases for milling as the radial depth of cutting increases when the feed rate is fixed for all assisted cutting methods. Similarly, the cutting force increased for milling as the feed rate increased when the radial depth of cutting is fixed for all assisted cutting methods, as shown in Figure 9.

The cutting force increases as the feed rate and radial depth of cut increase; the effective cutting area grows, more chips are formed per unit time, and the contact arc length between the tool and the workpiece increases as the cutting depth increases. When the radial depth of cutting is 0.10 mm, and the decreasing trend of cutting force is not obvious. Due to the deep radial depth of cutting, the effective laser heating depth may be insufficient, and the feed rate will also affect the laser heating temperature, resulting in the declining trend in cutting force being not readily visible due to the inadequate preheating influence. Therefore, this sub-section shows that the cutting force is proportional to the feed rate and the radial depth of cutting.

In total, 81 experiments were performed (27 experiments for each assistance system). The average cutting force values of three experiments are reported in Table 7.

### 3.3. Edge Morphology Comparison

After the silicon nitride milling experiments are completed, the silicon nitride material is properly cleaned and placed on a tool microscope to observe the edge morphology of the silicon nitride with a 20x objective lens. There are many pictures, so only a few more relevant edge shapes are selected for comparison. The edge morphology of milling using three kinds of assistance are shown in Figure 10. After comparing the three kinds of auxiliary processing, it can be found that unassisted and laser-assisted processing will have defects such as edge chipping, fluffing, and material pull-out regardless of any process parameters. Reducing the cutting force, but not enough to soften the silicon nitride, cannot effectively reduce machining defects. It can be clearly seen from Figure 10 that unassisted milling has the highest edge defect and hybrid-assisted milling has lowest edge defect. When the feed rate is 1500 mm/min and the radial depth of cutting is 0.10 mm, due to the faster feed and deeper cutting depth, the laser preheating time on the machined surface is reduced and the preheating depth is insufficient, resulting in the highest edge defect seen on silicon nitride, as shown in Figure 10b. However, when the laser is combined with triaxial ultrasonic assistance, the improvement in edge morphology is immediately apparent. It should be noted that the periodic separation of the ultrasonic vibration tool and the workpiece minimizes extrusion and friction between the workpiece and the tool. Due to the effect of triaxial high-frequency vibration, the triaxial ultrasonic assistance combines the advantages of rotary ultrasonic and biaxial ultrasonic, so that the silicon nitride matrix is vibrated to form finer chips during grinding, to prevent material from being scraped off by the PCD end mill, resulting in a better surface roughness.

### 3.4. Tool Wear Comparison

The milling cutter was removed and cleaned after the silicon nitride experiments were done, and the tool wear was evaluated using a tool microscope. The milling cutters and grinding rods utilized in the experiment, as well as the Taguchi method of the three auxiliary milling and grinding tests, are compared in this sub-section. The studies employed a total of 27 milling cutters. Each experiment is repeated three times with one cutter, for a total of 81. Also, we investigated the impact of various factors on the tool. From Figure 11, it can be seen that tool wear of hybrid-assisted milling is significantly better than that of laser-assisted milling and unassisted milling. At the feed rate 1500 mm/min and radial depth of cutting of 0.1mm, serious chip build-up and chipping defects can be clearly observed, and with the increase in the feed rate, it can be seen that the tool wear had a worsening trend. It is speculated that when the depth of cutting is deep and the feed rate is fast, the preheating time of the laser on the machined surface is reduced and the preheating depth is not enough, which cannot effectively reduce the bonding strength of the material and increase the cutting load of the tool. Adhesion to the tool leads to poor chip evacuation, resulting in a reduction in the sharpness of the edge of the milling cutter, and the tool is affected by the cutting load in this state, resulting in more serious edge cracking of the milling cutter. When the laser is combined with the three-axis ultrasonic assistance, the intermittent motion generated by the high-frequency vibration of the three-axis allows the air to enter the cutting area smoothly, so that when the tool interacts with the workpiece, the silicon can be directly sheared on the tool. A breaking phenomenon occurs, and the chips are effectively discharged, reducing the chipping of the cutting edge of the milling cutter, and greatly improving the life of the tool. However, although the hybrid-assisted method operates at a feed rate of 1500 mm/min and the radial depth of cutting is 0.1mm, the cutting edge is still cracked. When the depth of cutting is deep, the feed is fast, and the ultrasonic amplitude is maintained at the same size, the chip accumulation on the cutting edge will be aggravated, and the tool will be affected by this state. The impact of ultrasonic high-frequency vibration causes the silicon nitride chips on the tool to collide with the workpiece again, causing the edge of the milling cutter to slightly crack.

### 3.5. Results of the Validation Experiment

The results of the ANOVA analysis were validated through an experiment. A short overview of the findings of the validation experiments are presented here. The findings of the experiment are expressed in terms of cutting force values, surface roughness, edge morphologies, and tool wear (see Figure 12).

The experimental findings confirm the results of the ANOVA analysis. The validation experiment shows the surface roughness values for unassisted, laser-assisted, and hybrid-assisted systems to be 0.612, 0.323, and 0.234 µm, respectively, while the cutting force values for these three assistance systems are found to be 18.22, 6.8, and 4.3 N, respectively.

## 4. Conclusions

This work aims to perform edge milling tests on silicon nitride using unassisted, laser-assisted, and hybrid-assisted (three-axis ultrasonic plus laser). Surface roughness, cutting force, edge morphology, and tool wear are all used to validate the differences, advantages, and disadvantages of each auxiliary effect. The following conclusions are drawn from the above analysis and discussion:The impact of each variable on the surface roughness of various auxiliary surfaces is investigated using variance analysis. The largest factor affecting the surface roughness of grinding is the feed rate, followed by the radial depth of cutting, and finally, the spindle speed; the largest factor affecting the surface roughness of milling is the feed rate, followed by the radial depth of cutting, and finally the spindle speed.Compared with no assistance, the use of laser and ultrasonic assistance can obtain better surface roughness and edge morphology for milling. The material processed by the milling cutter has better surface roughness. However, PCD milling cutters have a shorter tool life. The surface roughness of silicon nitride increases as the feed rate and radial depth of cutting increase. Under the complete Taguchi experiment, when compared to unassisted milling, laser-assisted and hybrid-assisted milling improved the total average surface roughness by 42% and 66%, respectively.The cutting forces are high for milling. The cutting force increases as the radial depth of cutting and feed rate increase for both milling and grinding. When compared to unassisted milling, laser-assisted and hybrid-assisted milling lowered the total cutting forces by 26% and 46%, respectively.The use of dual-axis laser and triaxial ultrasonic waves can effectively reduce sur-face roughness, cutting force, and edge morphology while improving tool life.The Taguchi method optimized parameters in this experiment plan are high spindle speed, low feed rate, and low radial depth of cutting.The validation experiment confirmed the findings of the ANOVA analysis.

These experimental findings will be helpful in the parameter planning and machining assistance of silicon nitride end milling and can actually improve the surface accuracy of the workpiece and improve the quality. The following are suggestions for future research and development:

The amplitude of the ultrasonic wave and the power of the dual-axis laser are crucial factors. So, these should be investigated in the future at different temperatures.

In the future, in order to find the position of the carbon dioxide laser, the laser path simulation program should be adopted, and the coaxial red light should be used, to make the dual-axis laser-assisted method more complete.

## Figures and Tables

**Figure 1 materials-16-00137-f001:**
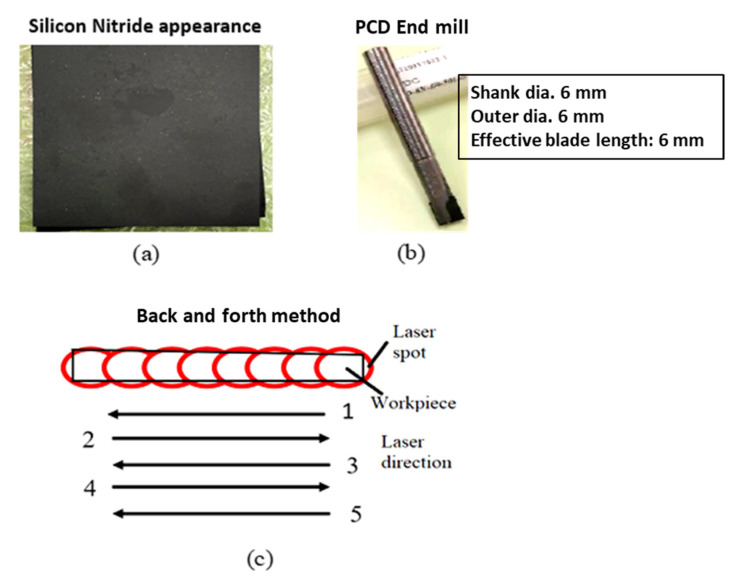
(**a**) Silicon nitride appearance, (**b**) PCD milling cutter, and (**c**) schematic illustration of the back-and-forth method.

**Figure 2 materials-16-00137-f002:**
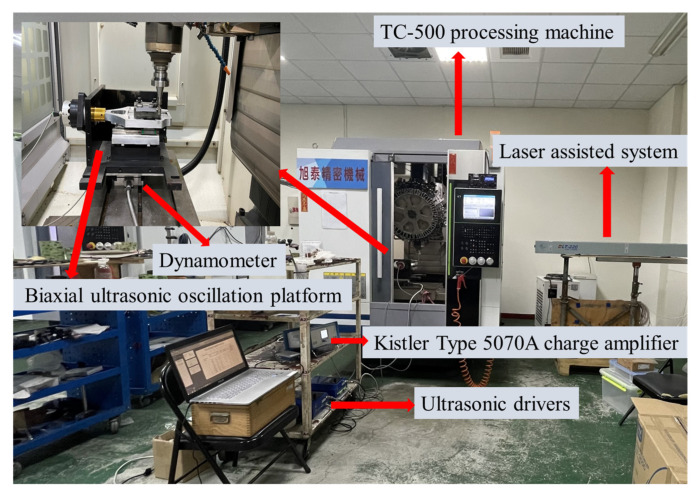
Description of the experimental setup.

**Figure 3 materials-16-00137-f003:**
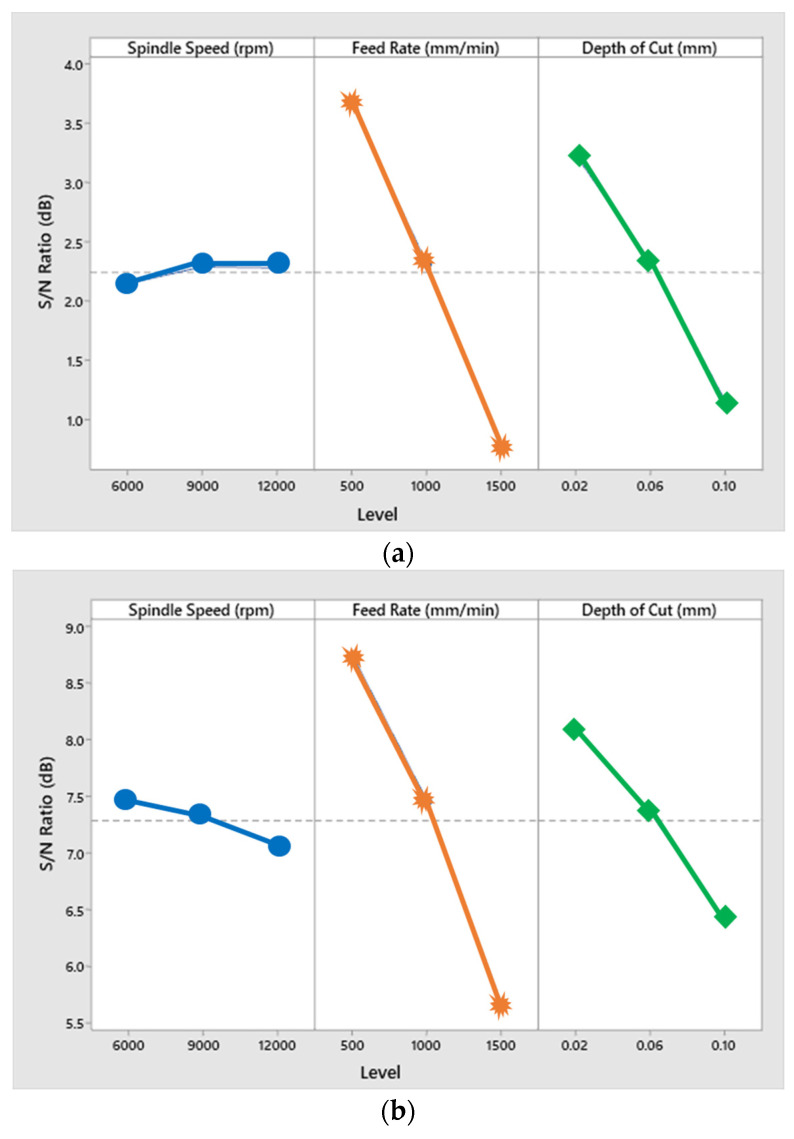
Surface roughness factor response plot for (**a**) unassisted, (**b**) laser-assisted, and (**c**) hybrid-assisted systems.

**Figure 4 materials-16-00137-f004:**
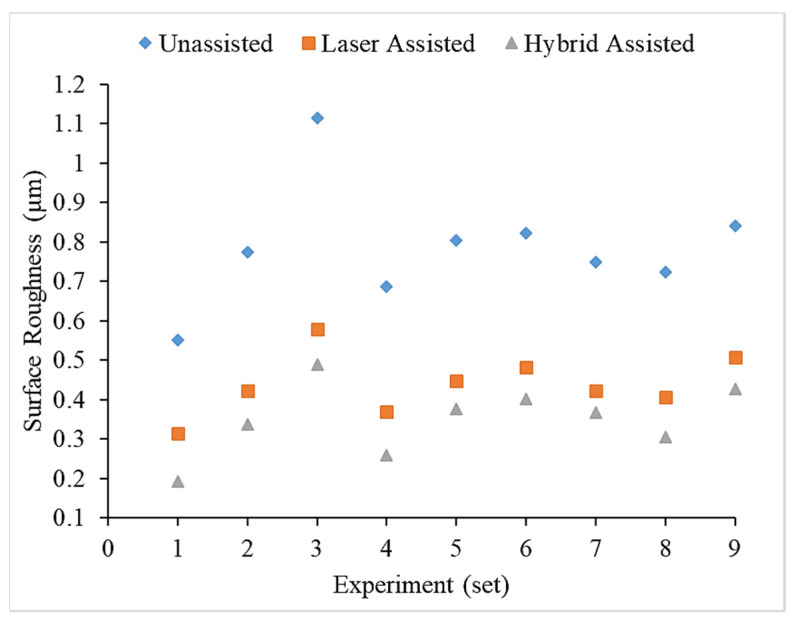
Surface roughness comparison chart with the three kinds of assistance using the Taguchi method.

**Figure 5 materials-16-00137-f005:**
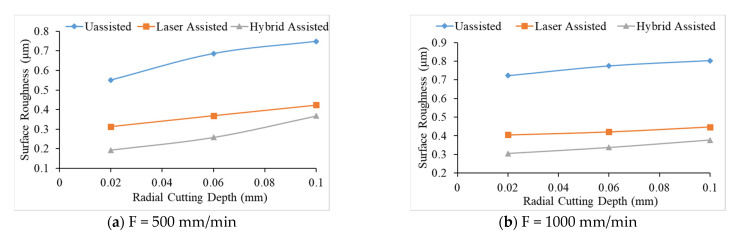
The relationship between radial depth of cutting and surface roughness under different feeds for unassisted, laser-assisted, and hybrid-assisted systems.

**Figure 6 materials-16-00137-f006:**
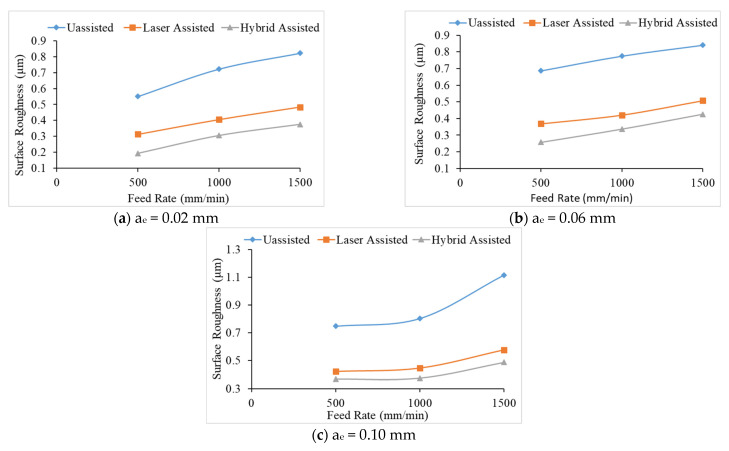
The relationship between feed rate and surface roughness under different radial cutting depths for unassisted, laser-assisted, and hybrid-assisted systems.

**Figure 7 materials-16-00137-f007:**
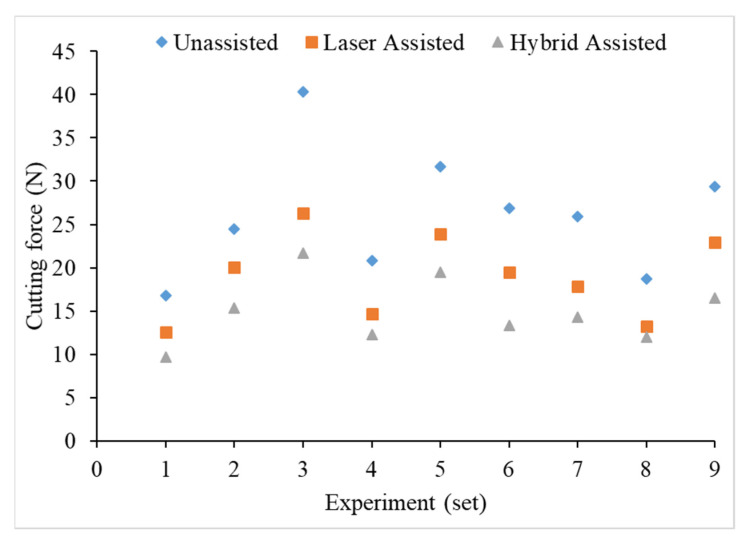
Comparison chart of the cutting force of various process parameters of the Taguchi method for unassisted, laser-assisted, and hybrid-assisted systems.

**Figure 8 materials-16-00137-f008:**
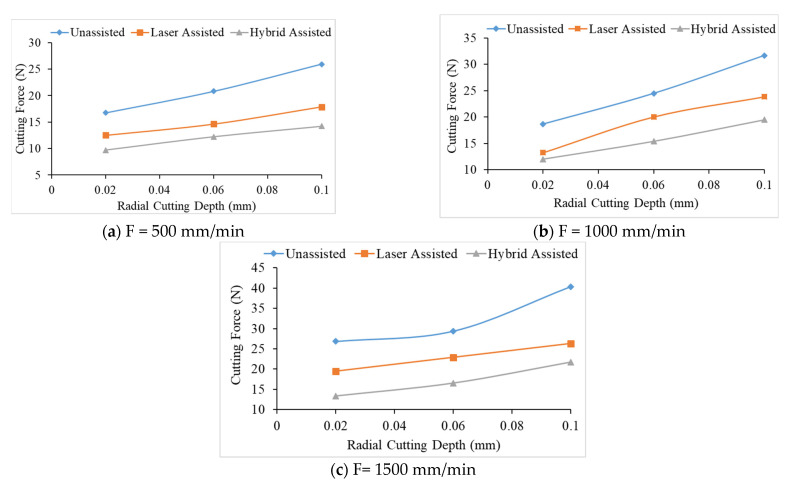
The relationship between the radial depth of cutting and cutting force under different feeds for unassisted, laser-assisted, and hybrid-assisted systems.

**Figure 9 materials-16-00137-f009:**
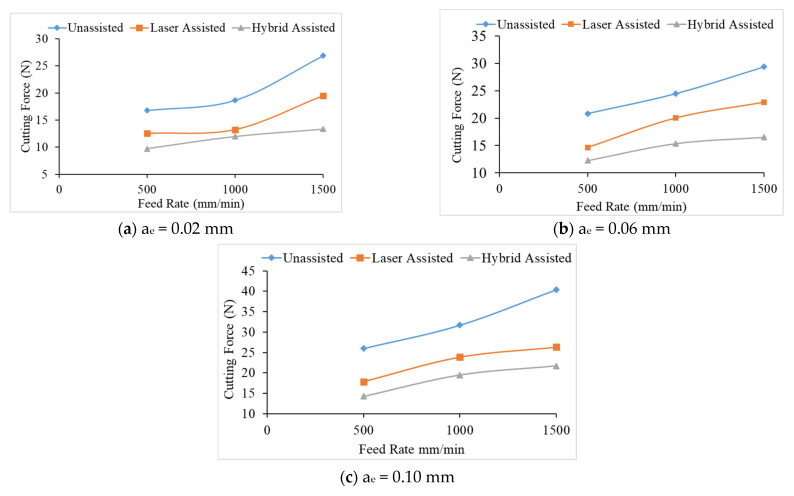
The relationship between the feed rate and cutting force under different the radial depths of cutting for unassisted, laser-assisted, and hybrid-assisted systems.

**Figure 10 materials-16-00137-f010:**
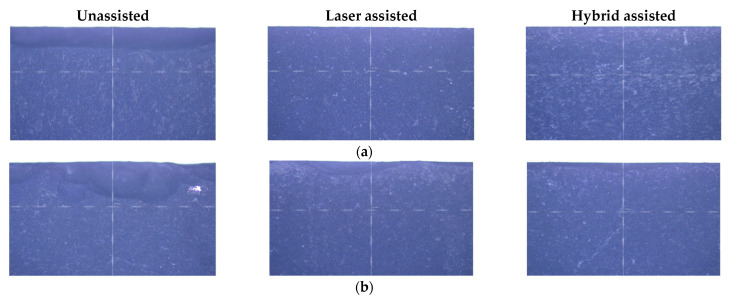
Edge morphology of silicon nitride milling at (**a**) n = 6000 rpm, F = 500 mm/min, and a_e_ = 0.02 mm; (**b**) n = 6000 rpm, F = 1500 mm/min, and a_e_ = 0.10 mm; (**c**) n = 9000 rpm, F = 1000 mm/min, and a_e_ = 0.06 mm; and (**d**) n = 12,000 rpm, F = 1000 mm/min, and a_e_ = 0.02 mm.

**Figure 11 materials-16-00137-f011:**
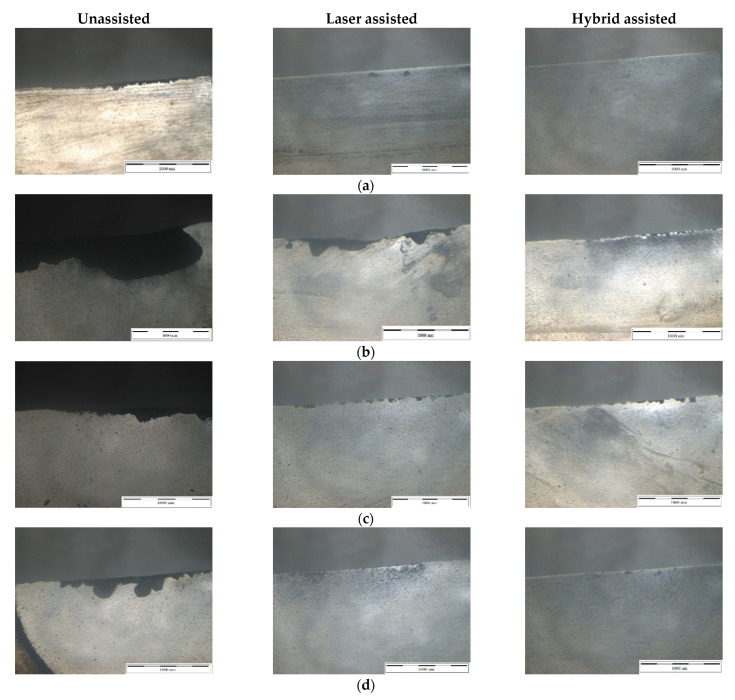
Surface morphology of milling cutters at (**a**) n = 6000 rpm, a_e_ = 0.02 mm, and F = 500 mm/min; (**b**) n = 6000 rpm, a_e_ = 0.10 mm, and F = 1500 mm/min; (**c**) n = 9000 rpm, a_e_ = 0.06 mm, and F = 1000 mm/min; and (**d**) n = 12,000 rpm, a_e_ = 0.02 mm, and F = 1000 mm/min.

**Figure 12 materials-16-00137-f012:**
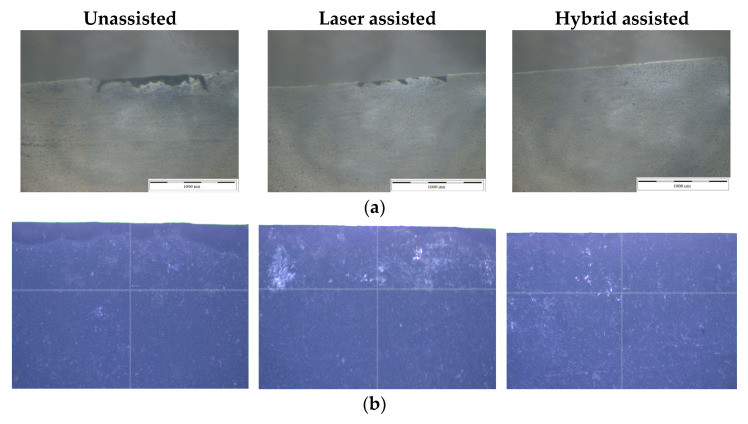
Validation experimental results at n = 12,000 rpm, a_e_ = 0.02 mm, and F = 500 mm/min for (**a**) surface morphology and (**b**) edge morphology for unassisted, laser-assisted, and hybrid-assisted systems.

**Table 1 materials-16-00137-t001:** Silicon nitride mechanical and physical properties.

Characteristics	Unit	Numerical Value
Density	g/cm^3^	3.2
Hardness	HV	1700
Bending strength	MPa	700
Fracture toughness	MPa m^1/2^	4.5
Elasticity modulus	GPa	320
Melting temperature	°C	1900

**Table 2 materials-16-00137-t002:** L9 orthogonal parameter planning table.

Grinding rod diameter d (mm)	Φ6 PCD end mill
Axial depth of cut a_p_ (mm)	2
Spindle speed n (rpm)	6000, 9000, 12,000
Radial depth of cut a_e_ (mm)	0.02, 0.06, 0.10
Feed rate F (mm/min)	500, 1000, 1500

**Table 3 materials-16-00137-t003:** Silicon nitride milling L9 orthogonal array.

	Experimental	Spindle Speed (rpm)	Feed Rate (mm/min)	Radial Depth of Cut (mm)
Group Parameters				
1		6000	500	0.02
2		6000	1000	0.06
3		6000	1500	0.10
4		9000	500	0.06
5		9000	1000	0.10
6		9000	1500	0.02
7		12,000	500	0.10
8		12,000	1000	0.02
9		12,000	1500	0.06

**Table 4 materials-16-00137-t004:** Unassisted milling (Si_3_N_4_) surface roughness variance analysis.

Factor	Level (S/N)			Sum of Square (SS)	Contribution Rate (*ρ*%)
	1	2	3		
A(n)	2.148	2.290	2.2811	0.0378	0.17
B(f)	3.657	2.312	0.750	12.270	59.85
C(a_e_)	3.228	2.330	1.160	6.451	30.41
Error				2.016	9.50
Total				21.2084	100

**Table 5 materials-16-00137-t005:** Laser-assisted milling (Si_3_N_4_) surface roughness variance analysis.

Factor	Level (S/N)			Sum of Square (SS)	Contribution Rate (*ρ*%)
	1	2	3		
A(n)	7.455	7.325	7.075	0.224	1.25
B(f)	8.741	7.453	5.661	14.355	73.76
C(a_e_)	8.087	7.358	6.40	4.245	21.81
Error				0.636	3.17
Total				19.461	100

**Table 6 materials-16-00137-t006:** Hybrid-assisted milling (Si_3_N_4_) surface roughness variance analysis.

Factor	Level (S/N)			Sum of Square (SS)	Contribution Rate (*ρ*%)
	1	2	3		
A(n)	9.983	9.408	8.803	2.090	4.36
B(f)	11.580	9.427	7.188	28.942	60.45
C(a_e_)	10.847	9.542	7.805	13.967	29.18
Error				2.879	6.01
Total				47.878	100

**Table 7 materials-16-00137-t007:** The average cutting force values of 27 milling experiments.

Experiment	Unassisted (N)	Laser Assisted (N)	Hybrid Assisted (N)
1,2,3	16.77	6.53	2.7
4,5,6	24.48	14.02	8.38
7,8,9	40.37	20.3	14.73
10,11,12	20.82	8.63	5.25
13,14,15	31.68	17.85	12.49
16,17,18	26.85	13.5	6.35
19,20,21	25.96	11.85	7.25
22,23,24	18.66	7.23	4.98
25,26,27	29.39	16.9	9.54

## Data Availability

Correspondence and requests for materials should be addressed to Shen-Yung Lin.

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
