# Peer review of "Experimental Investigation of Milling Performance of Silicon Nitride Ceramic Subject to Different Assisted Systems"

_materials, 2022, doi:10.3390/ma16010137_

Round 1

Reviewer 1 Report

Manuscript Number: Materials-2059458

Title: Experimental Investigation of Milling Performance of Silicon Nitride Ceramic Subject to Different Assisted Systems

Decision: Major revision

Article Type: Article

I think it should be reviewed and missing information should be added before it is published in the journal. The article is, in general, well written but there are some issues that article should consider to revise in order to improve its quality. Some comments were done in this way:

Ø  Fig.2 quality is very poor. Also let's take a closer picture and please see the details in fig. Let's explain on it.

Ø  “For statistical analysis, this article employs the experimental design approach pioneered by Dr. Genichi Taguchi in the 1950s. The experiment plan is based upon Taguchi method L9 orthogonal table.” No need for this sentence, please remove it.

Ø  Let's give all the features of the cutting tool used.

Ø  Let's directly give the experimental parameters (6000, 9000, 12000) instead of the level values (1,2 and 3) given in Table 3.

Ø  Take Table 4 from the program as it is and copy it. Also, do not place it in the table. S / N ratios should be given.

Ø  The S/N response table must be given directly.

Ø  The Anova table should be given directly.

Ø  Lütfen, Fig.3’de yer verdiÄŸiniz a,b ve c grafiklerini küçültüp birlikte görülebilecek hale getiriniz.

Ø  For the spindle speed, feed rate and depth of cut in Fig.3, please select a different marker shape and color for each of them from within the program.

Ø  The originals of the values given in Fig.9 should be given for 27 experiments by reducing the “The record of data” image.

Ø  The quality is very low in Fig.10. There is no scale. Surfaces are incomprehensible. Image should be taken with another device (may be SEM).

Ø  With which device were the cutting tool wears given in Fig.11 taken. Cutting tool wear amounts should be measured. It should be charted and the wear comments should be evaluated on this chart. It is necessary to find out what percentage of the effect of the experimental conditions and interpret the wear over these values. In addition, the BUE and chipping defects reported on lines 272 and 273 must be proven by EDS analysis. You can benefit from the following studies in the analysis of these wear and BUE images.

·        https://doi.org/10.1016/j.jmapro.2020.08.034

·        https://doi.org/10.1016/j.jmrt.2020.01.010

·        https://doi.org/10.3390/ma7031603

·        https://doi.org/10.3139/120.111571

·        10.1088/2631-7990/abc26b

In addition, these publications will add innovation to your literature.

Ø  Expressions such as slightly cracked lines in lines 291 and 292 should be avoided. These should be measured and interpreted numerically with the data.

Ø  A thread can be opened for anova results done before the results. Here, the surface roughness and wear rate of Spindle Speed, Feed rate, Radial depth of cut should be mentioned. Then, which parameter gives the optimum result, these values should be interpreted.

Ø  No confirmation experiment was performed after Anova analysis. I think that it will be a more remarkable publication if the best results after taguchi and anova are estimated and supported by a validation experiment.

After making the above corrections would recommend this article for publication in Materials.

Reviewer 2 Report

Please find the attached review comments.

Reviewer 3 Report

I gave it a couple of times and read still its unclear to me what is the novelty. Is it based on the existing cutting tool modification or the improvement of another side? also how this can connect to UNs SDG goals? Your work should satisfy at least one SDG.

Use elaborative captions for your Figures.

From Figure 2 it seems several patterns of graphs are possible. What types of data you have collected from the PC?

Figure 3: feed cut one seems to experience no changes. More analysis is required for this phenomenon. 

How the data for figure 10 was collected? if you want to connect Figures 2 and 10 how this is possible? Reading this figure is quite hard. this Blue background is natural or any other colour can be employed?

also in terms of reference use of 19 journals is quite less.  improve this part too

Round 2

Reviewer 1 Report

The authors have made the desired corrections for the article. If the editor wishes, the article can be published in the journal as it is.

Reviewer 3 Report

accept